# Layered Weighted Blended Order-Independent Transparency

Fabian Friederichs*
TH Köln

Martin Eisemann†
TU Braunschweig

Elmar Eisemann‡
Delft University of
Technology

## ABSTRACT

Our approach improves the accuracy of weighted blended order-independent transparency, while remaining efficient and easy to implement. We extend the original algorithm to a layer-based approach, where the content of each layer is blended independently before compositing them globally. Hereby, we achieve a partial ordering but avoid explicit sorting of all elements. To ensure smooth transitions across layers, we introduce a new weighting function. Additionally, we propose several optimizations and demonstrate the method's effectiveness on various challenging scenes in terms of geometric- and depth complexity. We achieve an error reduction more than an order of magnitude on average compared to weighted blended order-independent transparency for our test scenes.

**Index Terms:** Computing methodologies—Computer graphics—Rendering—Rasterization;

## 1 INTRODUCTION

Real-time rendering of semi-transparent objects is challenging for a rasterizer. The classic approach is costly, as it sorts primitives or objects by depth and alpha-blends them back-to-front or front-to-back, which heavily depends on the scene's depth complexity. Further, intersecting primitives or objects are difficult to handle. Order-independent approaches use depth-sorted fragments per pixel via A-Buffers [5] or depth peeling and its variants [4, 9]. Peeling costs can be reduced by a constant factor rendering several layers per pass [12] or by applying bin depth peeling [11] which introduces potential artifacts due to collisions within a bin. These depth peeling solutions do not scale well with depth complexity, reducing the framerate in scenes containing many layers of transparency, e.g. hair or smoke.

Current fast techniques approximate the correct solution. Easy-to-implement and resource-efficient are the blended order-independent transparency (OIT) operators [14]. They modify the classic alpha blending to make it order-independent at the cost of correctness. Unfortunately, they cannot always represent depth cues adequately, especially in scenes with many rather opaque surfaces (e.g. foliage) or different colors (e.g. colored smoke).

To address the blending accuracy, we extend the OIT algorithm [14] via a layering approach. We divide the view frustum into multiple depth intervals on which the weighted blended OIT algorithm is applied separately. Then the intermediate values are blended front-to-back to obtain the final result. This leads to more robust depth cues and improves color precision.

In the following section, we recap the existing order-independent blending operators (Sect. 2), introduce our algorithm (Sect. 3), including two variants and their implementation details (Sect. 4) before discussing results (Sect. 5) and concluding (Sect. 6).

---
*e-mail: fabian.friederichs@th-koeln.de
†e-mail: eisemann@cg.cs.tu-bs.de
‡e-mail: e.eisemann@tudelft.nl

## 2 RELATED WORK

The seminal Porter-Duff algorithm blends two semi-transparent colors $C_a$ and $C_b$ with opacity values $\alpha_a$ and $\alpha_b$ respectively: $C_{dst} = (\alpha_a C_a + (1 - \alpha_a)\alpha_b C_b)/(\alpha_a + (1 - \alpha_a)\alpha_b)$. Extending it to more than two surfaces requires ordering the surfaces, which is costly for modern GPU rasterizers. A plethora of methods has been developed to achieve or approximate transparency on a pixel basis, we refer to [17] for a good overview and focus on the most relevant.

Meshkin et al. introduced the "weighted sum" operator [15], which is basically an alpha blending ignoring order-dependent parts of the expanded expression, expressed as: $C_{dst} = \left(\sum_{i=1}^n C_i \alpha_i\right) + C_{bg}\left(1 - \sum_{i=1}^n \alpha_i\right)$, where $C_{bg}$ denotes the background color, $n$ the number of semi-transparent fragments in front of it. This operator works well with surfaces of small coverage but becomes increasingly inaccurate for $\alpha$-values $\geq 0.3$. Blending surfaces with large coverage values leads to colors outside of the span of the interpolants, i.e., causes over-darkening or over-brightening of the final color.

Replacing all occurrences of surface colors and coverage with their respective coverage-weighted averages, no longer ignores parts of the alpha-blending equation, while maintaining order independence:

$$C_{avg} = \frac{\sum_{i=1}^n C_i \alpha_i}{\sum_{i=1}^n \alpha_i}; \quad \alpha_{avg} = \frac{\sum_{i=1}^n \alpha_i}{n}$$
$$C_{dst} = C_{avg}\left(1 - \left(1 - \alpha_{avg}\right)^n\right) + C_{bg}\left(1 - \alpha_{avg}\right)^n$$

This removes the overdarkening issue of Meshkin's operator [15], but two other issues remain. Firstly, because every surface receives the same average coverage, invisible surfaces (zero coverage) can erroneously affect others [14]. Secondly, there is still no depth information incorporated, so it is impossible to tell if a surface is behind or in front of another surface.

McGuire and Bavoil [14] extended this approach by a pre-pass to determine the coverage $\alpha_{net} = 1 - \prod_{i=1}^n (1 - \alpha_i)$ to compute $C_{dst} = C_{avg}\alpha_{net} + C_{bg}(1 - \alpha_{net})$. They also incorporated depth by weighting contributions of fragments w.r.t their distance from the camera (Eq. 1). The weights used come from a monotonic decreasing function $w(z, \alpha)$, which has to be carefully tuned for each scene in order to provide good results (Eq. (7)–(10) in [14]).

$$C_{dwavg} = \frac{\sum_{i=1}^n C_i \alpha_i w(z_i, \alpha_i)}{\sum_{i=1}^n \alpha_i w(z_i, \alpha_i)} \tag{1}$$

Replacing $C_{avg}$ with $C_{dwavg}$ leads to the final weighted blended order-independent transparency operator (WBOIT).

$$C_{dst} = C_{dwavg}\alpha_{net} + C_{bg}(1 - \alpha_{net}) \tag{2}$$

The algorithm can have issues with rather opaque surfaces, particularly for dense clusters of transparent geometry with large distances between. Figure 1 shows a stack of squares rendered using the WBOIT technique. The material is amost fully opaque, i.e $\alpha \approx 1.0$. Hence, the frontmost, red square should occlude the others but the close proximity leads to a wrong result using the suggested depth-weighting functions [14]. To achieve better results, the ratio $\frac{w_i}{w_{i+1}}$ for two subsequent layers has to be very large. Rescaling the weighting function helps with accuracy but unfortunately one looses the ability to accurately combine surfaces farther away by doing so.

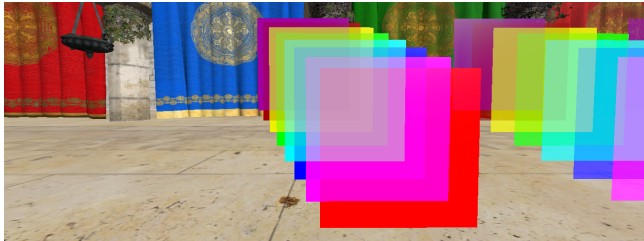

Figure 1: Inaccurate rendering of almost opaque squares due to an inadequate depth weighting function in weighted blending order-independent transparency operator.

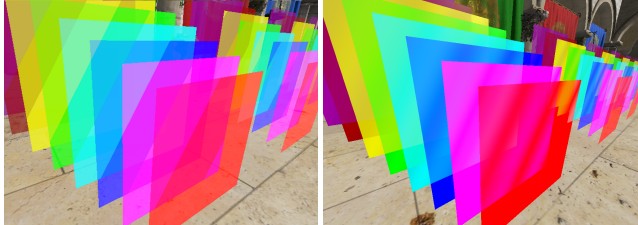

Figure 3: Left: Visible layer transitions where bin boundaries cut through primitives. Right: Smooth layer transitions improve the results but artifacts are still visible.

## 3 LAYERED WEIGHTED BLENDED ORDER-INDEPENDENT TRANSPARENCY

Figure 2 illustrates our basic algorithm. The view frustum is divided into a fixed number of depth intervals ("bins"). After applying WBOIT to every bin, the per-bin results are blended front-to-back using the *under* blending operator as described in [4]. Explicit sorting is avoided entirely and, instead, the bins enforce a partial order. After describing this basic solution (Sec. 3.1), we will increase its robustness, make it temporally coherent, and improve its accuracy (Sec. 3.2–3.4).

To further motivate our approach let us assume non-identical depth of different fragments per pixel. Let $z_\Delta$ be the minimum depth distance between two fragments of the same pixel, and let $z_{\min}$, $z_{\max}$ be the depth of the closest and furthest bin boundary respectively. If the number of bins $n$ fulfills $n > (z_{\max} - z_{\min})/z_\Delta$ the result of our algorithm is equal to the reference solution. If two fragment have the same depth value, our algorithm still produces correct results, due to the same depth weights computed by the WBOIT algorithm. We show an empirical evaluation of the influence the number of bins have on the overall error in Sec. 5.

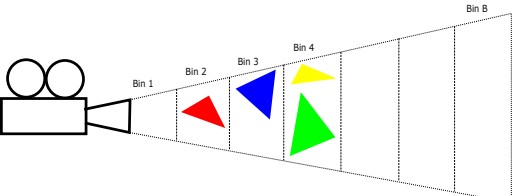

Figure 2: The view frustum is divided into a number of depth intervals or "bins". The WBOIT result is computed for every bin and the bin's results are combined front-to-back using the *under* operator. Here, we see a best case, where each primitive lies in one bin, generally, primitives can span several bins, which our full algorithm addresses.

### 3.1 Naive layering

For the naive approach, we apply OIT per bin $k$ (cf. Eq. (1) and ((2)), with $F_k$ denoting the list of fragment indices in the $k$-th of $B$ bins:

$$C_{dwavg_k} = \frac{\sum_{i \in F_k} C_i \alpha_i w(z_i, \alpha_i)}{\sum_{i \in F_k} \alpha_i w(z_i, \alpha_i)}, \quad \alpha_{net_k} = \left(1 - \prod_{i \in F_k} (1 - \alpha_i)\right)$$

The individual results are then combined using under blending [4] to discard fragments when the accumulated visibility of the background approaches zero. Given the opaque background $C_{bg}$, the

final pixel color is given by $C_f = A_{dst_B} C_{bg} + c_{dst_B}$, where

$$c_{dst_0} = \begin{pmatrix} 0 \\ 0 \\ 0 \end{pmatrix}, \quad A_{dst_0} = 1$$

$$c_{dst_k} = A_{dst_{k-1}} \left(C_{dwavg_k} \alpha_{net_k}\right) + c_{dst_{k-1}}, \quad A_{dst_k} = (1 - \alpha_{net_k}) A_{dst_{k-1}}$$

This naive approach is accurate if every primitive falls into a single bin (Figure 2). However, when bins cut through primitives layer artifacts can become visible (Fig. 3 left).

### 3.2 Smooth layer transitions

To reduce the visibility of layer boundaries (Fig. 3 right) we distribute the contribution of each rendered fragment to multiple bins using a bin weighting function. Its peak is at the bin's center and it decreases smoothly with increasing distance. A tent function seems natural but produces visible artifacts due to being only $C^0$ continuous. We chose a bell-shaped function instead:

$$b_k(z) = e^{-\left(\frac{\sigma}{(z_{bstart_k} - z_{bend_k}) \cdot 0.5} \cdot \left|\left(z_{bstart_k} + \left(z_{bend_k} - z_{bstart_k}\right) \cdot 0.5\right) - z\right|\right)^2},$$

where $z_{bstart_k}$ being the depth of the $k$-th bins near-plane $k$, $z_{bend_k}$ the depth of the respective far-plane and $z$ the depth value the function is evaluated for. Its peak is at the center of a bin with a value of 1.0 and it decreases smoothly to 0.5 at the bin's boundaries with $\sigma = 0.832555$ .

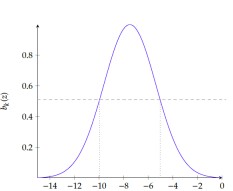

Instead of considering only the fragments in bin $k$, we now iterate over all fragments $F$ of a pixel and weigh their color contribution with the bin weight function $b_k(z)$:

$$C_{dwavg_k} = \sum_{i \in F} C_i \alpha_i w(z_i, \alpha_i) b_k(z_i) / \left(\sum_{i \in F} \alpha_i w(z_i, \alpha_i) b_k(z_i)\right). \quad (3)$$

Additionally we must weigh the fragments' contribution to the coverage in the bin: $\alpha_{net_k} = 1 - \prod_{i \in F} (1 - \alpha_i b_k(z_i))$

### 3.3 Normalization

The multiplication of coverage $\alpha_i$ with the bin weight $b_k(z_i)$ in equation (3) introduces some error in the total coverage per bin $\alpha_{net_k}$. If there are only a few layers of transparent geometry near a bin boundary, the final color of that bin tends to get slightly too dark, because of premultiplied colors usage. It leads to visible layer artifacts, especially at shallow viewing angles (Fig. 3 right). For correct results the total coverage should be equal to $\alpha_{net}$ from the WBOIT algorithm.

To mitigate this problem, we employ a pre-pass, similar to [14], to compute the correct net coverage of the background. We further

include a normalization factor during blending of the bin's results. Under blending then simply becomes another weighted sum:

$$A_{dst_0} = 1, \quad C_{dst_B} = \frac{\sum_{k=1}^{B} C_{dwavg_k} \alpha_{net_k} A_{dst_{k-1}}}{\sum_{k=1}^{B} \alpha_{net_k} A_{dst_{k-1}}} \quad (4)$$
$$A_{dst_k} = (1 - \alpha_{net_k}) \cdot A_{dst_{k-1}}, \quad c_{dst_B} = C_{dst_B} \alpha_{net}$$

Note that, due to the normalization, the resulting color is not in premultiplied format anymore. Therefore, we multiply color $C_{dst_B}$ with coverage $\alpha_{net}$ before blending the result with the scene background.

### 3.4 Shaping the layers

Using depth maps can reduce artifacts for semi-transparent layers with great success [7, 18]. In another pre-pass, we render the depth of the first layer of transparent geometry. Using these depth values, we can offset the bins at each pixel, which increases precision and eliminates the artifacts for the first layer (Fig. 4).

The bin boundaries can still cut through the geometry of the remaining layers but these artifacts are typically strongly reduced. Furthermore, the final color is invariant w.r.t view-space depth since layers and depth weighting function are not moving directly with the camera anymore. Hereby, one issue of the original WBOIT algorithm is fixed. A dependency on the viewing angle, which has an impact on the depth distance between layers remains (Fig. 5 and 6) but are less noticeable. Furthermore, if all primitives share the same orientation, all layering artifacts are removed (Fig. 7).

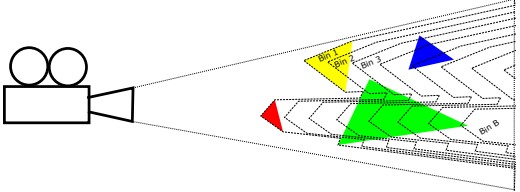

Figure 4: The shape of the layers match the first layer of transparency.

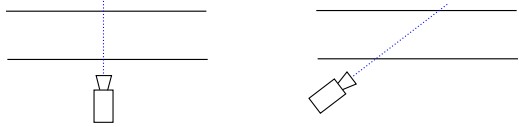

Figure 5: The depth distance between two bins changes with viewing angle.

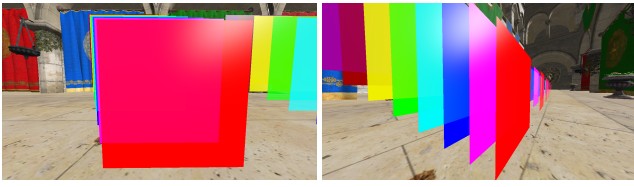

Figure 6: Influence of the viewing angle on the final color.

We additionally shift the origin of the weighting function towards the camera to prevent unreasonable depth weights for the first layer, Fig. 8. This option allows to trade off between accuracy in highly opaque surfaces and accuracy in fairly transparent surfaces using the amount of shift as a single parameter.

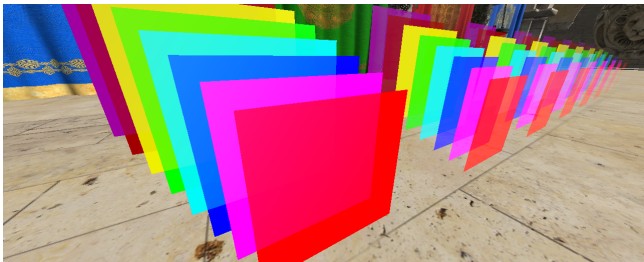

Figure 7: Adaptive layers remove visible layering artifacts.

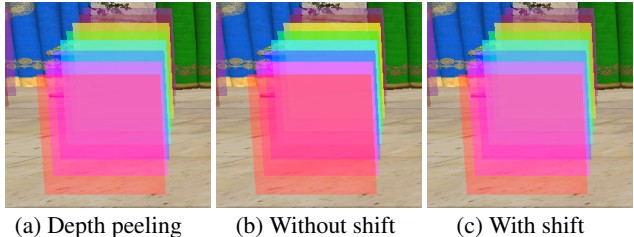

(a) Depth peeling     (b) Without shift     (c) With shift

Figure 8: Shifting the depth weight function towards the camera improves results compared to ground truth. Notice the red tint in (b). MSE are (b) 602.44 and (c) 12.49 .

## 4 STORING AND DISTRIBUTING FRAGMENTS

To implement our rendering method, we will discuss two implementations to store the rendered fragments: per-pixel linked lists and layered rendering. The first will collect all fragments that will virtually be associated with a bin in a postprocess, the second enables us to collect the content directly in bins. The linked lists method is advantageous in scenes with high geometric but lower depth complexity, while layered rendering is beneficial for scenes with high depth complexity but lower geometric complexity.

Per-Pixel Linked Lists    This implementation uses per-pixel linked lists to store the rendered fragments before blending them using our approach from Sec. 3. While advanced schemes for per-pixel linked lists exist that are cache efficient and robust regarding capacity issues, we chose for an easy-to-implement approach storing color, depth and an index to the next fragment [2]. This solution works well when the manually-set capacity (usually a multiple of the target resolution) is not exceeded. If the buffer's capacity is reached, remaining fragments are discarded. Consequently, complex scenes might require a high memory consumption and can lead to potential buffer overflow artifacts.

To produce the final image, we apply the layer shaping (Sec. 3.4) implicitly. The final rendering pass without geometry loops over all collected fragments $f$ per pixel, normalizes their position, finds each respective bin $b_f$, applies the corresponding blending weights $w_f$, and accumulates the result in an array of bin values $v[b_f]$. Finally, the values in $v$ are blended to determine the final transparent color that is blended with the background.

Layered Rendering    One can render into multiple layers of a 2D-array texture or 3D texture simultaneously with layered rendering. Each layer represents a bin and the primitives are redirected to the bins based on their spanned depth interval from within the GPU's geometry shader. In practice, geometry-shader instancing allows us to invoke the shader a user-defined number of times per primitive. As 32 instances are guaranteed on current hardware [10], we render batches of 32 bins per pass, adding additional passes if more than 32 bins are requested, and blend the result batch by batch.

To produce the final image, the span of each triangle is detected and if the instance intersects the depth interval of the corresponding

bin, we output the triangle via layered rendering. We apply the weights corresponding to the bin and the fragments position in the fragment shader. In a final render pass without geometry, we blend the different layers together.

Unfortunately, when layer shaping is used, the span of a triangle would need to be determined via the depth map. NBuffers [6] and Mipmaps can be used to compute an estimate by extracting the min/max value over a bounding square but this adds an overhead to the method. In practice, we emit the triangle to all layers and resolve the span in the fragment shader. For a small amount of layers, one can also use multiple render targets (current GPUs typically support eight) to avoid geometric duplications.

## 5 EVALUATION

We evaluated our approach on seven scenes and compared to the original WBOIT algorithm [14] and depth peeling [9] as a reference solution (Fig. 12). The scenes all feature different depth- and geometric complexity and are summarized in Table 1. The distribution of non-opaque fragments is highly non-uniform in image space and depth (Fig. 9). This challenges our algorithm, as a uniform distribution would be best for performance. Table 2 shows the tested algorithms and abbreviations used in the diagrams. All measurements were conducted on an AMD Ryzen 7 3700X CPU with an 8GB AMD Radeon RX 5700XT at a $1280 \times 720$-pixel resolution.

| Scene | GC | DC | | | | Source |
|---|---|---|---|---|---|---|
| | | per pix. | | per bin | | |
| | | Ø | max. | Ø | max | |
| squares | 128 | 0.86 | 8 | 0.01 | 2 | BG from [13] |
| dragon high poly | $8,713k$ | 0.57 | 22 | 0.01 | 8 | [13] |
| powerplant | $12,748k$ | 3.23 | 95 | 0.05 | 31 | [13] |
| colored smoke | $\sim 2,100$ | 6.53 | 320 | 0.13 | 99 | BG from [13] |
| grey smoke | $\sim 120k$ | 361.11 | $18.7k$ | 7.14 | $13.8k$ | BG from [13] |
| burning wood | $225k$ | 7.66 | 76 | 0.10 | 50 | [16] |
| engine | $150,674$ | 2.61 | 87 | 0.03 | 18 | [1] |

Table 1: Test scenes. GC: Geometric Complexity (num. non-opaque triangles), DC: Depth Complexity (avg./max. num. non-opaque fragments per pix and bin), BG: Background. The number of bins per scene is listed in Fig. 12.

| Abbreviation | Algorithm |
|---|---|
| DP | Depth Peeling |
| WBOIT | Weighted Blended Order-Independent Transparency |
| SNDLL$\{8, 16, 32, 64\}$ | Smoothed, normalized, depth buffer offset per-pixel linked lists |
| SNDLR$\{8, 16, 32, 64\}$ | Smoothed, normalized, depth buffer offset layered rendering, |

Table 2: Abbreviations of tested algorithms. The numbers in curly braces denote bin counts.

We measured the average rendering time for transparent geometry and blending with the opaque background (Fig. 11). Mean squared error with respect to the depth-peeling reference is reported globally and per-pixel in a heatmap (Fig. 12). For qualitative comparisons, we chose the highest number of layers/bins, while maintaining real-time performance ($> 25$ fps) on our system with all optimizations enabled. We also analyze the performance and mean squared error with respect to the number of bins (Fig. 10).

### 5.1 Results

We provide performance measurements (Fig. 9–11) and visual comparisons (Fig. 12). Our baseline algorithms perform as expected

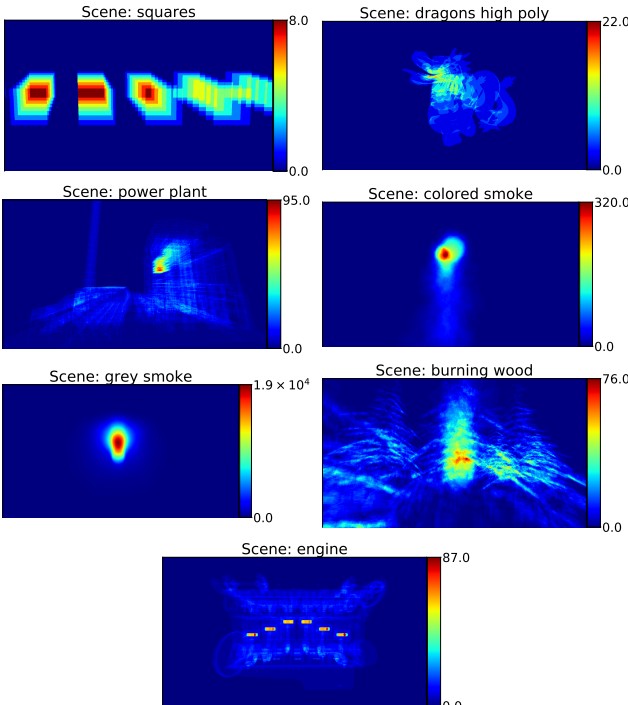

Figure 9: Heatmaps showing the depth complexity per pixel for our test scenes. The scaling has been normalized for each scene for better visualization.

with WBOIT, a very simple two-pass algorithm, being the fastest technique in all cases at the cost of reduced quality and DP being the slowest with highest quality.

Bin count: Fig. 10 plots the effect of the bin count vs. MSE (top row) and rendering time (bottom row) exemplarily for two test scenes (the results for all test scenes are available in the supplemental material). We see for most scenes an exponential reduction in the MSE and a clear convergence towards the reference solution which reinforces our theoretical considerations from Sec. 3. Small irregularities in convergence can appear as neighboring fragments in depth might or might not fall into the same bin, depending on the number of bins. The only exception is the *grey smoke* scene as the number of bins becomes irrelevant with high depth complexity but low color variance.

Looking at the bin count vs. frametime we can see a few interesting patterns in the data. In geometrically complex scenes, such as "power plant", the linked list approach works extremely well and the limiting factor here is the rendering itself. On the other hand the layered rendering approach struggles to deliver good performance in these scenes as multiple rendering passes are required. The sudden "jumps" in the performance are due to the limit on the number of rendering targets that can be rendered during one pass. Scenes with a low geometric complexity, such as "grey/colored smoke" and "burning wood", show a clear linear increase in rendering time with respect to the number of bins which is desired as it shows that no bigger additional overhead is produced by our algorithm.

Squares: Very low geometric and depth complexity. Our linked-list versions outperform layered rendering due to the overhead of the geometry-shader stage. Increasing the bin count from 32 to 64 leads to a significant increase in frame times because two 32-batches of geometry shader invocations are now needed instead of just one. Color accuracy using our approach is improved reducing the MSE compared to WBOIT by a factor of $46\times$.

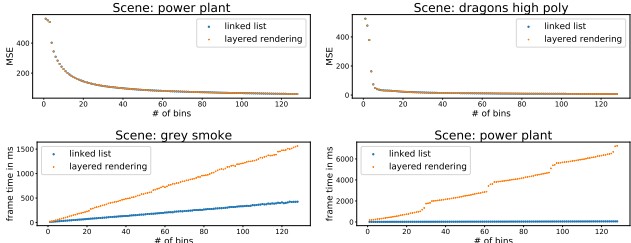

Figure 10: Top row: Influence of bin count on the MSE. Bottom row: Influence of bin count on the rendering time.

**Dragon high poly:** High geometric and low depth complexity. The linked list approach delivers excellent performance and again improves color accuracy compared to WBOIT by a factor of 8.6×. The layered rendering approach is not applicable for such geometrically complex scenes and is even slower than depth peeling.

**Power Plant:** High geometric and medium depth complexity. Again the layered rendering approach comes to its limits due to the complex geometry. The linked list approach performs well and reduces the MSE compared to WBOIT by a factor of 5.6×.

**Colored smoke:** Low geometric and high depth complexity. Here, layered rendering outperforms the linked lists. The linked lists are flooded with fragments and the mostly random memory access becomes a bottle-neck. Since the particle billboards all face the camera, the primitives do not span a large depth range, making discarding in the geometry shader effective. This is a tough case for WBOIT in terms of visual quality. The red particles appear to be even in front of the blue ones. Color accuracy improves by a factor of 16.4× compared to WBOIT.

**Gray smoke:** Low geometric, very high depth complexity. The linked list buffers are flooded with fragments and most are discarded. Depth peeling needs around 3 minutes to peel all layers of a single frame. Layered rendering benefits again from the primitive discard optimization but the variants with depth buffer offset struggle due to the high amount of geometry. The scene is optimal for WBOIT because all particles share a uniform color, which hides blending artifacts. Still, color accuracy improves by a factor of 1.05× compared to WBOIT.

**Burning wood:** Medium geometric and depth complexity. We included this one to test rendering of foliage, which combines very opaque and very transparent objects. WBOIT attributes an insufficient contribution to the nearby needles with respect to the orange particles, creating the impression of needles disappearing. This leads to clearly visible artifacts around the stem. Layered rendering produces more accurate images improving the results by a factor of 1.3× compared to WBOIT.

**Engine:** Medium geometric and depth complexity. This scene is typical for CAD design, where surfaces are often semi-transparent. Layered rendering suffers when using a too low number of bins (here 16). WBOIT cannot cope well with the different materials encountered, resulting in a higher overall error but also strong error spikes in several positions. The linked lists work well in this scene reducing the MSE compared to WBOIT by a factor of 4.4×.

## 5.2 Discussion

Our layering approach enhances color accuracy and can improve depth cues. When applying layer shaping, artifacts are eliminated to a large extent and it partially solves the problem of color variations when the distance to the camera changes. However, a slight dependence on the viewing angle is introduced, which can be visible for very simple and consistent shapes, such as the squares. Organic

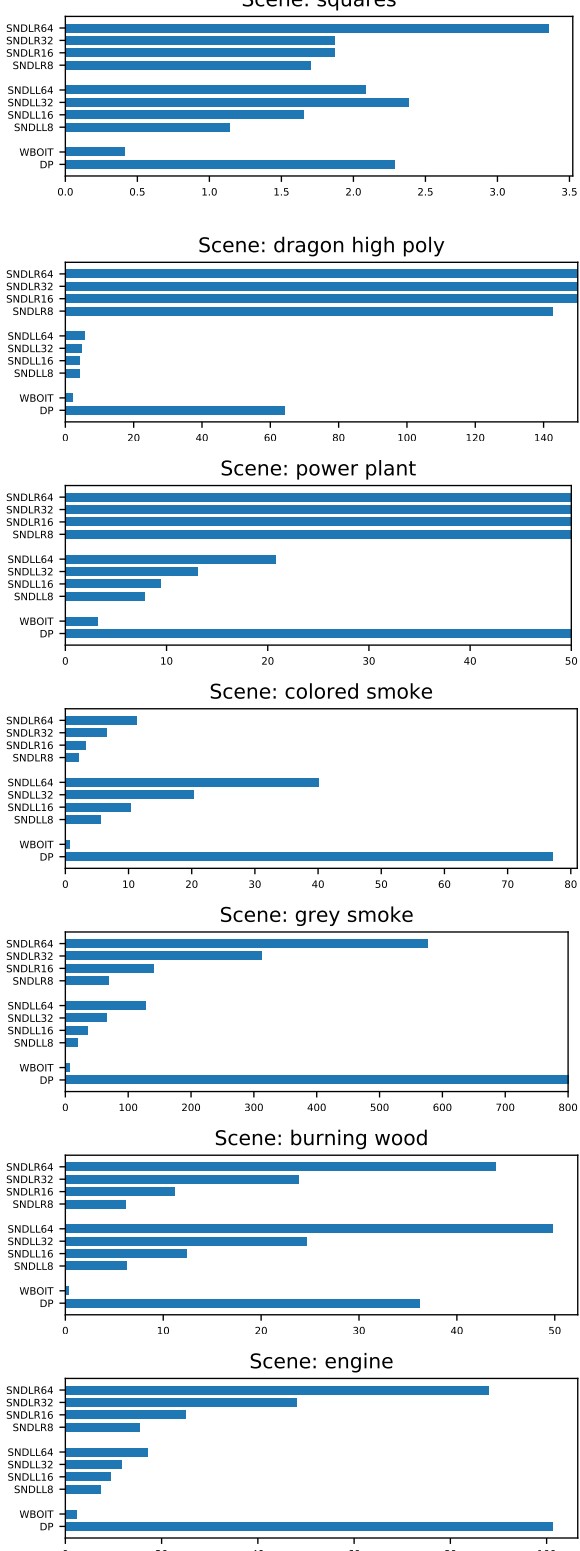

Figure 11: Performance measurements (frame times in ms).

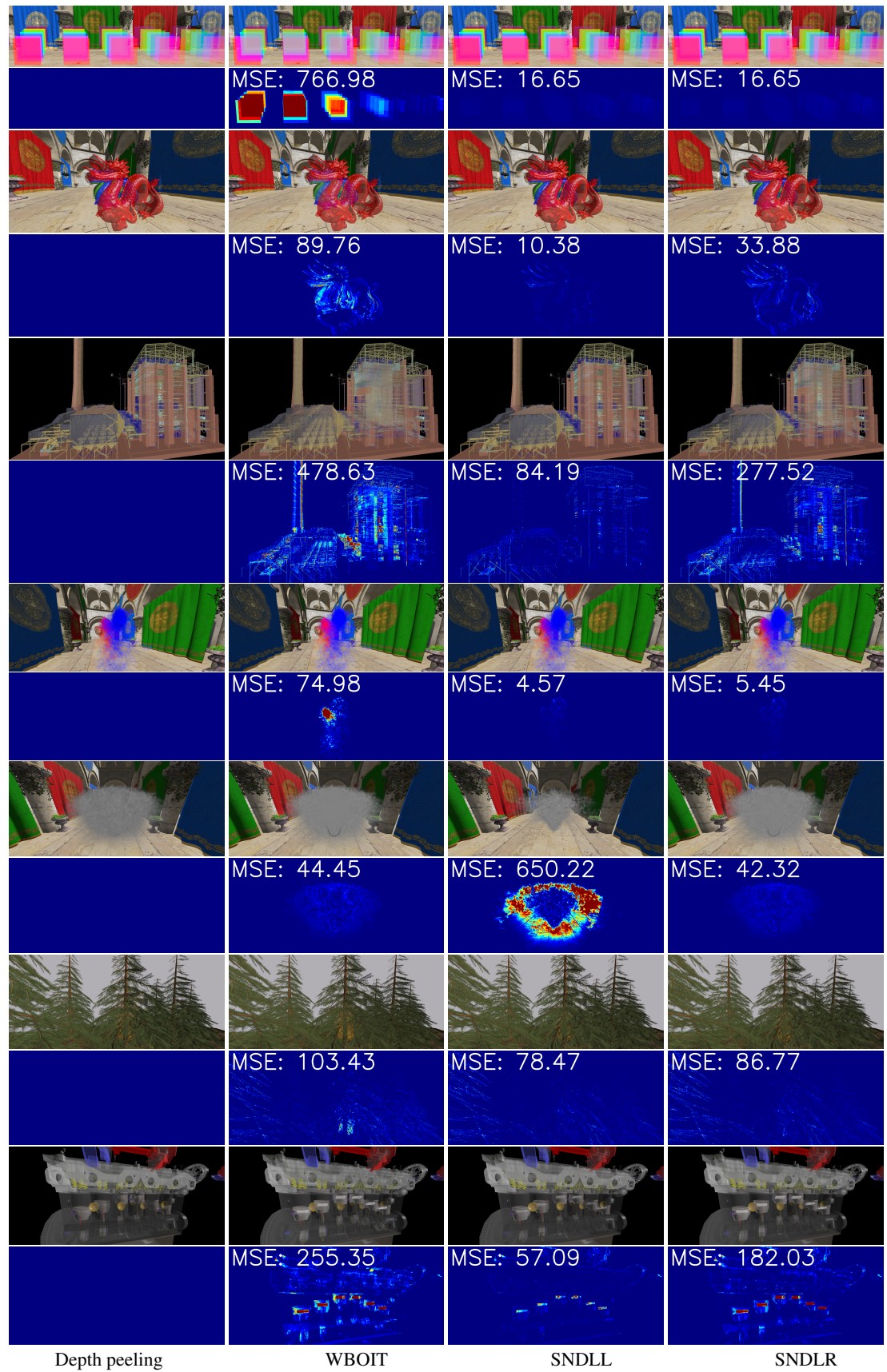

Figure 12: Visual comparison. Left to right: Depth peeling, WBOIT, our methods. Odd rows: rendered images, even rows: error images. Top to bottom: "squares" (with 64 layers both for SNDLL and SNDLR), "dragon high poly" (64 for SNDLL and 8 for SNDLR), "power plant" (64 for SNDLL and 8 for SNDLR), "colored smoke" (64 for SNDLL and SNDLR), "grey smoke" (16 for SNDLL and 8 SNDLR), "burning wood" (32 for SNDLL and 32 for SNDLR), "engine" (64 for SNDLL and 16 for SNDLR)

shapes, like the Stanford Dragon, are less susceptible to this effect and most layering artifacts are hard to perceive.

The linked-list approach works well for complex transparent geometry but runs into memory issues for a high depth complexity. Using layered rendering introduces an overhead for complex geometry but delivers good performance for high depth complexity because only a portion of the fragments must be held in memory at any given time. High geometric *and* depth complexity remains a challenge (Table 3). Still, our algorithm handles several scenes that are problematic for existing algorithms. For example, it renders particle systems well, which rely heavily on sorting, as well as detailed transparent objects, which would typically result in a serious reduction of quality.

| | Geometric Complexity | |
|---|---|---|
| Depth complexity | Low to Medium | High |
| Low to Medium | LL / LR | LL |
| High | LR | ? |

Table 3: The linked list (LL) and layered rendering (LR) distribution methods favor different kinds of scene configurations.

## 6 CONCLUSIONS

We introduced a layered solution of weighted blended order-independent transparency that improves color accuracy. We achieved smooth layer transitions by introducing a bin weighting function and shaping the bin decomposition based on the first layer of semi-transparent surfaces. Our approach is easy to implement and builds upon two variants; per-pixel linked lists and layered rendering. It avoids sorting of fragments but enforces a partial order implicitly, the latter enabling results closer to a reference solution. Our algorithm shows good performance for challenging cases, such as particles with high depth but relatively low geometric complexity, or scenes with only a few layers of transparency but rather complex geometry and therefore closes a gap in existing OIT algorithms.

In the future, we want to test techniques such as K-Buffers [3] or stochastic transparency [8] to bound the memory requirements and optimize the list structures, e.g., by coupling them to screen tiling for a reduced memory consumption, increased throughput and improved cache friendliness.

## ACKNOWLEDGMENTS

This work was partially funded via the NWO Vernieuwingsimpuls VIDI Grant NextView. The authors would like to thank 3D Warehouse [1],TurboSquid [16], and McGuire [13] for providing the test scenes.

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
