# OpenReview forum: "Layered Weighted Blended Order-Independent Transparency"
_graphicsinterface.org/Graphics_Interface/2021/Conference — GI 2021_

### Official Review · AnonReviewer1 · 2021-01-05
**Layered Weighted Blended Order-Independent Transparency**

**Rating:** 7
**Confidence:** 3

**Review:**


This paper extends order-independent transparency using a layered approach in which
each layer is independently composed using Blended Order-Independent Transparency (WBOIT) before
being blended with other layers using the implicit sorting they provide. Blending artifacts are
tempered using hand-crafted depth-dependent coefficients. The paper presents results on
a large variety of scenes showing different levels of depth and geometric complexity.

This paper uses lots of "mathematical cooking": It is a bit disappointing that no theory
(as opposed to purely heuristic choices) backs up the use of particular equations, such as b_z or
values like \sigma.

Unless I'm mistaken, I haven't seen in the result section a study of the effect of varying parameters
such as the number of bins (I'm not saying layers because this term is used in Table 2 to
characterize scene complexity). I would really like to see, for a given scene, a curve showing the blending error
as a function of number of bins, as well as other parameters (such as sigma!) to see how the method
behaves. From the material that is given in the supplementary, creating such plots would be fairly straightforward.

This paper presents a rather simple idea but it is short enough to make it acceptable. I believe it would
make a valuable contribution to CGI.

Notes:
- end of 3.1: "This naive approach is accurate if...". It's not very accurate, but rather "visually acceptable".
- in Figure 8, it would help a lot to show a reference image. Furthermore, on paper it is pretty impossible to
distinguish the 3 images

---

### Official Review · AnonReviewer2 · 2021-01-12
**minor advance to classic OIT problem**

**Rating:** 6
**Confidence:** 3

**Review:**

This paper presents a method for order-independent transparency, a longtime problem in real-time rendering. This paper combines McGuire & Bavoil's weighted blended order-independent transparency with depth binning, dividing depth uniformly and applying WBOIT to obtain a color for each bin, then combining the bins for the final pixel color. The results are a middle ground in quality and speed between plain depth peeling and the WBOIT approximation. The comparison is less favorable if depth peeling acceleration techniques are used, but still there are potential use cases for the proposed method.

The paper offers a minor contribution to the classic OIT problem and on that basis can be considered for acceptance. One weak point is that the results may be a little overstated relative to the state of the art. There has been a fair amount of work on OIT and just referring to Wyman's survey is not sufficient to cover it. In particular, the strand of work on depth peeling acceleration has not been discussed in the present paper. Acceleration methods can easily halve the cost of plain depth peeling (Everitt 2001) and one of these would be a more fair comparison.

It is not clear how easy it would be for a graduate student not previously familiar with OIT methods to reproduce this work. Given the practical nature of this research, a bit more advice about implementation might be worthwhile.

I would have liked to see some additional characterization of the scenes, e.g., heat maps of layers (rather than just maximum layer
count), and some commentary on the number of layers per bin. It is easy to imagine that for some cases such as the foliage, most bins are empty and the work is being done by a few bins with most of the layers. It would also be worthwhile to show the relationship between error and bin count. The heat maps of error are quite helpful; it might also be instructive to zoom in on regions of particularly large error to show the improvement.

I was happy to see that the authors reported their results in ms per frame rather than attempting to report fps.

---

### Official Review · AnonReviewer3 · 2021-01-13
**The paper presents some accuracy improvements over order independent rendering of scenes with transparent objects, without requiring total back-to-front ordering of objects. A number of heuristics are introduced to get a practical solution. The contribution is more in the practive category.**

**Rating:** 6
**Confidence:** 4

**Review:**

The topic of this paper is rendering of scenes with transparent objects. Usually, transparent objects are rendered in back to front order, which requires sorting. Sorting is computationally expensive and slows down rendering of large scenes with high depth complexity. Order independent methods have been proposed but usually result in artifacts and inaccuracies. The authors present a method that avoids total sorting, while providing reasonably good approximation.  They slice the view frustum into  bins, and assign objects  to these bins. Bins are rendered in order. Objects within a bin are dealt with in an order independent manner. Since total sorting is avoided, computational complexity is reduced. A number of heuristics are introduced to make the method yield reasonable results, such as objects crossing bin boundaries, distance within a bin, etc. . They have experiments which show improvements over the order independent method.

The main contribution is in deciding to replace total ordering with partial ordering using bins. The method has its limitations, but seems practical as long as scenes do not have both high depth and high geometry complexity.

---

### Meta-Review · Area_Chair1 · 2021-01-16

**Recommendation:** Accept
**Confidence:** 4

**Metareview:**

All reviewers had a positive overall opinion of the submission.  Therefore acceptance is recommended.  Some small issues were identified and the authors are encouraged to address them in the revised version.

---

### Decision · Program_Chairs · 2021-01-16

Accept